# Is it Feasible to Use a Low-Cost Wearable Sensor for Heart Rate Monitoring within an Upper Limb Training in Spinal Cord Injured Patients?: A Pilot Study

**DOI:** 10.3390/bioengineering9120763

**Published:** 2022-12-03

**Authors:** Miriam Salas-Monedero, Vicente Lozano-Berrio, María-Jesús Cazorla-Martínez, Silvia Ceruelo-Abajo, Ángel Gil-Agudo, Sonsoles Hernández-Sánchez, José-Fernando Jiménez-Díaz, Ana DelosReyes-Guzmán

**Affiliations:** 1Biomechanics and Technical Aids Unit, Hospital Nacional de Parapléjicos (SESCAM), Finca La Peraleda s/n CP 45071, 14507 Toledo, Spain; 2International Doctoral School, Castilla La-Mancha University, 14507 Toledo, Spain; 3Nursing Department, Hospital Nacional de Parapléjicos (SESCAM), 14507 Toledo, Spain; 4Rehabilitation Department, Hospital Nacional de Parapléjicos (SESCAM), 14507 Toledo, Spain; 5Performance and Sport Rehabilitation Laboratory, Faculty of Sports Sciences, Castilla- La Mancha University, 14507 Toledo, Spain

**Keywords:** upper limb training, spinal cord injury, heart rate, wearable devices, biomedical application

## Abstract

(1) Background: Cervical spinal cord injury (SCI) patients have impairment in the autonomic nervous system, reflected in the cardiovascular adaption level during the performance of upper limb (UL) activities carried out in the rehabilitation process. This adaption level could be measured from the heart rate (HR) by means of wearable technologies. Therefore, the objective was to analyze the feasibility of using Xiaomi Mi Band 5 wristband (XMB5) for HR monitoring in these patients during the performance of UL activities; (2) Methods: The HR measurements obtained from XMB5 were compared to those obtained by the professional medical equipment Nonin LifeSense II capnograph and pulse oximeter (NLII) in static and dynamic conditions. Then, four healthy people and four cervical SCI patients performed a UL training based on six experimental sessions; (3) Results: the correlation between the HR measurements from XMB5 and NLII devices was strong and positive in healthy people (r = 0.921 and r = 0.941 (*p* < 0.01) in the static and dynamic conditions, respectively). Then, XMB5 was used within the experimental sessions, and the HR oscillation range measured was significantly higher in healthy individuals than in patients; (4) Conclusions: The XMB5 seems to be feasible for measuring the HR in this biomedical application in SCI patients.

## 1. Introduction

Recent epidemiological studies show that the prevalence of SCI in Spain is between 270 and 380 million per inhabitants. According to data from the Spanish National Institute of Statistics, 2.86% of the Spanish population was affected by SCI [1]. SCI of non-traumatic etiology has been the main cause of SCI (58%), followed by SCI of traumatic etiology (42%), thus breaking the tendency of the last ten years, during which the main origin was traumatic [2].

After a SCI, patients have an affectation in the autonomic nervous system in lesions above the T6 metameric level. For this reason, patients with cervical SCI present differences compared to the healthy condition in terms of the cardiovascular adaptation level during the execution of a therapy or effort [3]. Another consequence of cervical SCI is the impaired UL function, which affects the performance of activities of daily living (ADL). Thus, the main objective of the therapeutic programme is for patients to achieve the maximum level of independence in the performance of ADLs [4,5].

Therefore, some of the most widely used methods to evaluate training effectiveness in adults, children, and adolescents generally include the use of HR monitors or pulse oximeters [6,7] and the use of motion sensors such as pedometers or accelerometers [8,9], based on the movement (or acceleration) of the limbs, and the trunk is closely related to the global energy expenditure.

Longitudinal studies on HR in SCI have analyzed HR in the acute and chronic phases of the injury, paying attention to the PA level that the patient performs in order to improve the performance of ADL activities [10,11,12].

On the other hand, in relation to the medical equipment used for recording the HR, all these studies have used different equipment, such as Holters digital blood pressure systems to monitor pressures, which records for an approximate period of up to a week [11]; and a HR monitor with ECG signal, option of pulxioximetry, non-invasive pressure, with presentation of HR reader [12]. However, in other studies, wearable technology like watches with chest straps and a heart frequency reader has been used [10]. El-Amrawy et al. [13] showed that the new wireless technologies, such as blood pressure trackers and body sensor devices could have a great impact on health care systems and quality of life, and analyzed the accuracy and precision of 17 wearable devices for HR monitoring. Bent et al. [14] demonstrated that wearables also provide excess HR information during low-intensity physical activity, which could be a safety mechanism built into the devices to ensure that patients do not exceed their maximum HR during training. Other authors have analyzed different characteristics of sensors with the aim of optimizing a wider variety of applications [15]. This is especially important for clinicians, as they would be aware of HR measurements during the training when they perform workload intensity assessments in movement-based trainings.

There is evidence about the use of Xiaomi Mi band device, showing the best package and choice, taking into account the quality–price ratio [13]. Based on this previous study, the XMB5 was chosen for the present research. However, these devices, which are commercial, are being purchased for leisure purposes as a way to quantify daily physical activity and hours of sleep, without paying attention to the heart rate readings they provide. In this respect, special attention is needed when dealing with patients, especially neurological patients. From our point of view, the novelty of this study focuses on the intention to use this device, XMB5, in the context of a biomedical application following a methodology designed for patients with cervical SCI. The main objective is to analyze the feasibility of this wearable device for monitoring the HR in cervical SCI patients. To reach this objective, an experimental study was conducted with the aim of comparing the HR measured by means of XMB5 with those measured by a capnograph/pulse oximeter used in the clinical setting for the monitoring of patients, simultaneously. Then, the Xiaomi wearable device will be used in a biomedical application, within a proof-of-concept study, for monitoring the HR in healthy people and cervical SCI patients during the performance of UL trainings within the rehabilitation program.

The remainder of this paper consists of the following: Section 2 provides all the information about the methodology, describing the study design, the participants’ characteristics, the equipment used, the experimental setup and methodology for data acquisition, and the statistical analysis; Section 3 is related to the results obtained in terms of variables at starting and ending conditions and variables related to each experimental session; Section 4 includes the discussion of the results, and Section 5 is the study’s conclusion.

## 2. Materials and Methods

This section includes all the information regarding the experimental protocol applied in this research. For that purpose, subsections about the study design, the participants included in the study, the experimental setup and data acquisition, the outcomes variables, and the statistical analysis were included.

### 2.1. Study Design

The study performed was descriptive, responding to an observational design with an experimental phase.

### 2.2. Participants

In this study, 8 subjects participated; they were divided into two groups: a neurologically healthy group formed by 4 subjects and a group of 4 SCI patients, all of them with UL motor function impairments. All patients suffered a cervical SCI with a metameric level between C4 and C8, and AIS grade between A and D, as defined by the International Standards for the Neurological Classification of Spinal Cord injury [16], specifically the upper limb motor score of the right arm (UER). Patients who presented any vertebral deformity, joint restriction, surgery on any of the UL, balance disorders, dysmetria due to associated neurologic or orthopedic disorders, or visual acuity defects were excluded. The UER was obtained with the clinical staff assessment of the strength of five muscle groups of the dominant UL. Each muscle group can be evaluated between 0 (no function) and 5 (normal function), with a total of 25 points.

This study was carried out in the Biomechanics and Technical Aids Unit of the Hospital Nacional de Parapléjicos (Toledo, Spain). All patients signed an informed consent form before the study. The guidelines of the declaration of Helsinki were followed in every case, and the study design was approved by the local ethics committee. Subject demographics are provided in Table 1.

### 2.3. Experimental Setup and Data Acquisition

HR was monitored using the XMB5, which is considered a low cost-device among the wearable technology devices [13]. The characteristics of the XMB5 bracelet are as follows. The sensor placement is in the central hole, which is well adjusted for the comfortable use of the bracelet. Moreover, the HR sensor is kept in contact with the skin to allow for the taking of measurements. The bracelet is manufactured in silicon with several measures for adjusting the bracelet to different wrist sizes (Figure 1). The sensor has an ADI accelerometer for quantifying the physical activity by means of the steps taken or running performed during the day. A previous study analyzed the optoelectronic characteristics of a silicon light-emitting device [17].

In the present study, the application Mi Fit, running under Android, was used for monitoring the HR. Measurements obtained from XMB5 were compared to a positive control usually used in clinical setting for the monitoring of patients, the professional medical equipment Nonin LifeSense^®^ II (NLII), capnograph and pulse oximeter, with CE marking as medical device [18,19,20]. This system was simultaneously used on the same hand wearing the XMB5.

Firstly, the HR measurements from XMB5 were compared to those measured by the NLII system on a sample of four healthy people. Both systems were used simultaneously, and Nonin was placed on the same hand wearing the XMB5, namely the left one. For each participant, 18 readings were recorded for each tracker simultaneously while the participant stayed in a resting position and then while the right arm performed the UL movements required by a UL activity. During the recording, the participant was seated in front of a table and the left hand remained at rest.

Following the completion of this first study, the UL training was performed in four healthy subjects and four subjects with cervical SCI. All participants performed the proposed UL training with the dominant arm, determined from the Edinburg dominance test [21]. The complete UL training in each participant lasted two weeks, with three training sessions per week of a duration of 30 min. Each session was based on UL activities by means of Leap Motion Controller (LMC) performing the “Robot Assembly” and “Petal picking” applications available in the Playground of LMC.

The duration of each session and the number of sessions per week were established while taking into account the recommendations of the resistance work of the American College of Sports Medicine (ACSM) for people with SCI [22].

The first step was to fit the XMB5 fitness wristband to the wrist of the dominant arm [21].

Each participant performed both therapeutic interventions in front of a table that was adjustable in height, with the aim of normalizing the starting position for all the participants. The arm was placed against the trunk; the elbow was flexed 90° in a neutral pronation-supination position and with the hand resting on the edge of the table, with the palm opened in medial position. Then, the LMC device was placed on the table at 75% of the maximum UL reaching movement of the participant to avoid compensatory movements with the trunk.

All the participants performed the experimental sessions in their own wheelchair or sitting in a conventional chair. The participants rested their back firmly against the back of the chair with an angle between the backrest and the seat of 90–100°, and their feet on the footrest at a 90° angle.

During the sessions, the participants wore the XMB5 used for the HR and the monitoring of the number of UL movements. The XMB5 was synchronized to the Smartphone of the researcher responsible of the experimental session. The mobile application used in this study was My Fit.

### 2.4. Outcome Measures

To compare both systems of HR monitoring, simultaneous measurements were taken in four healthy subjects. For each participant, 18 readings were taken in both experimental conditions: the static, in the absence of UL movement and during the performance of UL movements. In these situations, the variables analyzed were: the difference between the magnitude of the HR readings from both XMB5 and NLII medical equipment to analyze the agreement level of the simultaneous HR readings. Moreover, the trend of these readings was analyzed by means of a correlation analysis.

Therefore, the variables registered in the UL training previously described were as follows. As the training program was composed of several experimental sessions, two kinds of variables were registered: (a) variables within each experimental session and (b) variables measured at the beginning and at the end of the training over all the participants, both the healthy ones and the SCI patients.

Thus, within each experimental session, the HR was registered for analysis: the HR in baseline, the maximum, the minimum, the oscillation range, the HR at the end of the experimental session and the difference between the readings at the end and the baseline. In relation to the UL performance, the repetition number for each activity was registered. Moreover, the effort perception level by means of Borg Scale [23] was analyzed.

In relation to the UL training, all the subjects were evaluated at the beginning of the training and at the end by means of the Krupp scale, with the aim of determining the fatigue level associated to the neurological disease [24], and the amount of UL movements during a complete day was registered for all the subjects. To obtain this measurement, the subjects wore the XMB5 wristband over two complete days each time.

### 2.5. Statistical Analysis

The statistical analysis was performed by the SPSS 17.0 for Windows software (SPSS Inc., Chicago, IL, USA). The clinical and demographic characteristics of the participants were analyzed by descriptive statistics, showing the results as the mean and standard deviation.

In relation to the validation study between the XMB5 and NLII professional equipment, the paired t-test and a correlation analysis was conducted for the static and UL movement conditions.

In relation to the UL training program, the Mann–Whitney U non-parametric test was applied to analyze the differences between the two groups analyzed (healthy subjects and patients with SCI). The variables analyzed with this method were the HR reading at baseline and at the end of the experimental session, and the difference between both readings; the maximum, minimum, and the heart rate oscillation range during the performance of the experimental session; the number of UL movements; and the perception of effort according to the Borg scale. In addition, possible differences in the number of repetitions in each series were analyzed. Moreover, at the beginning and end of the complete training program, the punctuation in the Krupp scale and the number of total UL movements were analyzed for all participants.

## 3. Results

This section describes the results regarding the correlation between the HR measurements from both devices, XMB5 and NLII, in Section 3.1 and the analysis of the last experimental session in Section 3.2.

### 3.1. Feasibility of Using XMB5 Wristband

In this subsection, the results about the comparison between the HR readings from the XMB5 wristband and NLII medical equipment are shown.

The Pearson correlation coefficient between the readings from both systems suggests a strong and positive linear association between both variables: r = 0.921 (*p* < 0.01) in the static condition and r = 0.941 (*p* < 0.01) for the HR readings taken during the performance of UL movements. This trend is shown in the scatter graphics included in Figure 2.

Both XMB5 and NLII systems registered similar HR readings in the static condition: 73.80 ± 4.79 and 73.97 ± 4.61, respectively. During the performance of UL movements, both systems also registered similar readings: 90.33 ± 9.38 for XMB5 and 90.20 ± 7.96 for Nonin medical equipment.

The differences observed between both XMB5 and NLII system were not statistically significant: t(71) = −0.754, *p* = 0.454 for the static condition, and t(71) = 0.323, *p* = 0.748 during the execution of UL movements.

### 3.2. Last Experimental Session within the UL Training

The sample analyzed was divided into two groups; neurologically healthy subjects and SCI subjects that were matched in age, weight, and height. Thus, no statistically significant differences were found for demographic characteristics between both groups analyzed (Table 1).

After performing the UL training based on the execution of six experimental sessions, the last session was chosen for analysis, because in that session, the healthy and SCI subjects all reached optimal performance.

Statistically significant differences were found between both groups (neurologically healthy and SCI) in variables related to the training performance: the number of repetitions in both activities within the training was lower in SCI subjects than in healthy people: in activity 1, 24.50 ± 12.25 and 11.00 ± 10.50 repetitions, respectively and in activity 2, 80.00 ± 34.25 and 26.50 ± 21.50, respectively, in each group. In relation to the HR, the statistical differences were found in the maximum value, in the oscillation range, and in the difference between the readings at baseline and at the end of the experimental session (Table 2).

Furthermore, it was observed that the HR before the session, in a baseline situation, was similar for both groups analyzed. However, after the experimental session, healthy subjects had a higher HR value than SCI subjects (101.00 ± 38.25 > 78.00 ± 36.00), but no statistically significant differences were found. In relation to the difference between the HR readings at the end and baseline conditions, statistically significant differences were found between healthy and SCI subjects (38.50 ± 37.25 > 9.00 ± 15.50, *p* < 0.05) (Table 2 and Figure 3).

Within the last experimental session, statistically significant differences were found between the two groups in the maximum value of heart rate and the oscillation range measured. The maximun value was higher in healthy people (149.50 ± 64.50) than in SCI patients (100.50 ± 20.75, *p* < 0.05) and, as a consequence, this difference was found in the oscillation range in healthy vs. SCI patients (92.50 ± 65.50 > 44.50 ± 14.25, *p* < 0.05).

The effort perception, analyzed through the Borg scale, was greater for the SCI patients group than for the healthy individuals, but no significant differences were found.

### 3.3. Variables at Baseline and at Ending the Complete UL Training

In this case, the variables related to the Krupp scale and the number of UL movements were analyzed, and the results are shown in Table 3.

Fatigue decreased after the end of the complete training in both groups analyzed. However, no significant differences were found within the groups between the two assessments: in healthy people (2.33 ± 0.61 > 1.99 ± 0.48) and SCI patients (3.38 ± 1.28 > 2.49 ± 0.69). On the other hand, in relation to the UL movements, there was an important increase for this variable in SCI patients between both assessments (1.418 ± 9.800 < 12.409 ± 5.496), expressed as thousand units (Figure 4).

## 4. Discussion

The results obtained in this study as a proof of concept suggest that the XMB5 wristband seems to be a feasible way to monitor HR in SCI patients. This device was chosen for this study because other authors previously found that this device was the best in relation to quality and price [13]. In relation to the materials, the wristband was manufactured in silicon with several measures for adjusting the wristband to different wrist sizes. This characteristic was very important in the context of a biomedical application because it allowed for the wristband to be placed on different patients without difficulty.

The ADI accelerometer was used for measuring the number of UL movements in SCI patients from the number of directional changes in the arm, considering that the wristband is placement on the wrist. Other commercial wristbands based on GPS for quantifying physical activity were not suitable for measuring small and low-level UL activity in patients.

Obviously these measurements have an estimation error. For that reason, the first step was the feasibility analysis of the HR readings from XMB5 against those simultaneously measured by a clinical professional device used for HR monitoring in patients (NLII). In this study, a difference between both systems around the mean HR value of 0.166 bpm in absolute value was obtained with a standard deviation of 1.876 bpm in the absence of movement with both arms in a rest situation. However, during the execution of the UL movements, the standard deviation measured was higher (3.284 bpm). Nevertheless, the correlation analysis between the HR readings of both systems was high. There is no evidence from similar studies against which to compare the results obtained in this research, but there are those who suggest the importance of such studies in a clinical setting to monitor HR for any treatment [14]. It is very important for many people with acute CML to ensure that they do not exceed their maximal HR during exercise, which is a circumstance that can lead to adverse cardiac events [25].

The measurements obtained during the undertaking of activities by wearable sensors may contribute more accurate and objective insights into clinically meaningful changes in impairment, activity, and participation during the neurological rehabilitation process [26]. Furthermore, the XMB5 wearable device has been used in a biomedical application, within a small experimental study, for HR monitoring in healthy people and SCI patients during the performance of UL training. It has been proven that wearable sensors give us an excess of information about HR during low-intensity physical activity, such as that which is carried out in the different treatments and therapies in the neurorehabilitation process, which could be a safety mechanism incorporated into the patients’ devices so that they do not exceed their maximum HR during the execution of therapeutic activities [14].

In this study, there was a decrease in the HR oscillation range between session 1 and session 6 in healthy subjects and people with SCI. This adaptation in healthy subjects is generally explained by an increase in the sympathetic pathway activity [27]. However, because people with cervical SCI have limited sympathetic efferent innervations to the cardiovascular system [28,29], studies suggest that changes in HR variability as soon as a the HR is low could be regulated by the parasympathetic branch [30]. This means that this adaptation to exercise should, in the case of SCI, lead to an improvement in the vagal modulation of the HR dynamics, related to the theory of HR variability “trainability” [31]. In fact, this study showed that people with cervical SCI are trained, since the HR readings at the end of the experimental session are higher than the readings at baseline condition. Although the sample size of the experimental groups is small and therefore the study has been conducted as a pilot study, for the sample analyzed, a different behavior was observed between healthy participants and SCI patients in the difference between the HR measured at the beginning of the session and at the end, showing a different cardiovascular behavior with training. This is reflected in Figure 3. Therefore, patients retain the ability to produce positive adaptations in autonomous cardiac regulation, coinciding with other studies focused on the health of people with cervical SCI, since it shows that moderate or vigorous physical activity can be scheduled in the short term [32].

Because the sympathetic pathway and the vagal modulation increase, the effort perception in patients with cervical SCI and healthy people decreases as the number of sessions progresses, leading to the performance of activities with lower energy expenditure. This lower energy expenditure results in an increase in the number of executions of the activities proposed. It should be noted that healthy subjects obtained a higher performance, but this is due to the fact that people with SCI have limitations in the execution of UL activities, in a greater extent depending on the injury level and the severity classified according the ASIA scale. Moreover, a practice effect has been observed across the experimental sessions focused on the UL training by means of reaching and grasping movements [33]. These movements are non-ballistic and in a closed loop; they allowed sensory information to arrive, obtaining a feedback in the performance of the activities, helping healthy subjects and patients with SCI to improve these movements through a planning of the movement, thus managing to store a “sensorial trace” [34] with which they could compare their future executions by reinforcing that movement, improving their effectiveness and increasing the UL movements number.

Finally, it is necessary to take into account the limitations of the present study. Firstly, there is a limitation in relation to the size of the sample analyzed because this is a feasibility study with the aim of obtaining preliminary results in relation to the use of a low-cost device for HR monitoring during the performance of UL trainings.

On the other hand, a comparison with other studies in the literature is difficult due to there being no evidence from similar studies. In other studies on SCI populations that aim to measure the HR, different seated conditions were analyzed [34]. Moreover, the lack of standardization in the experimental protocols and the functional tasks chosen complicate the comparison between studies. In fact, the HR oscillation ranges measured depend on the UL functional task chosen.

Furthermore, the patients sample analyzed was very heterogeneous. Thus, the patient less affected in relation to the UL function was classified as a metameric level injury C8 and AIS classification D (P3). This patient showed a general UL functionality similar to a healthy subject, whereas the functional deficit in relation to the dexterity and ability of the hand may appear in dexterity clinical tests. However, the results obtained in relation to the patient P4 were very different because the cervical injury was higher and more severe. In both patients, the HR oscillation range was lower than in healthy subjects because of the affectation in the autonomous nervous system.

Taking into account these aspects, a further study should be made with a greater sample of SCI patients. Therefore, an electromyographic registry should be made simultaneously with the aim of detecting the appearance of possible peripheral fatigue during the performance of UL training in SCI patients.

## 5. Conclusions

Low-cost technology, in this case XMB5, seems to be suitable for monitoring the HR during UL training by means of technology, LMC, in patients with cervical SCI. For this reason, this methodology will be incorporated within a biomedical application for registering the HR within an UL training based on immersive virtual reality applications for SCI patients with the aim of determining the optimal intensity and duration of the experimental sessions.

## Figures and Tables

**Figure 1 bioengineering-09-00763-f001:**
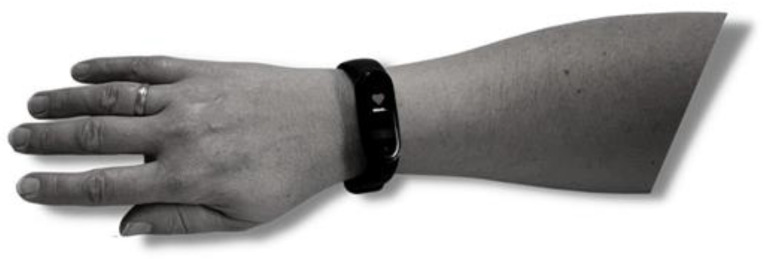
All participants wore the Xiaomi Mi Band 5 on their right arm during the experiments.

**Figure 2 bioengineering-09-00763-f002:**
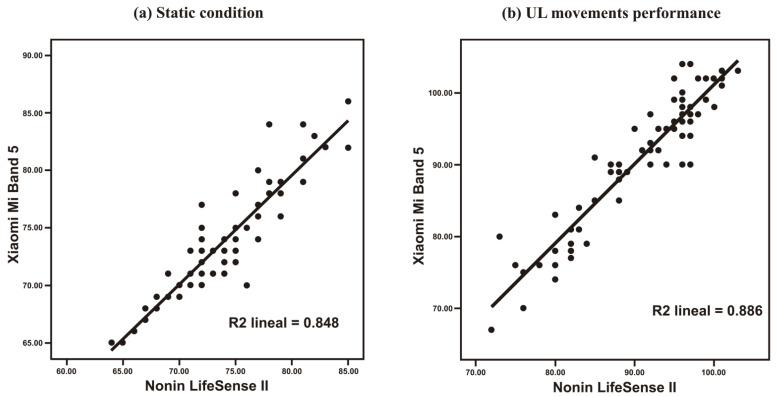
Scatter diagrams between the heart rate measured by Nonin Lifesense II and the heart rate measured by Xiaomi Mi Band 5 wristband during a static posture (**a**) and dynamic condition (**b**).

**Figure 3 bioengineering-09-00763-f003:**
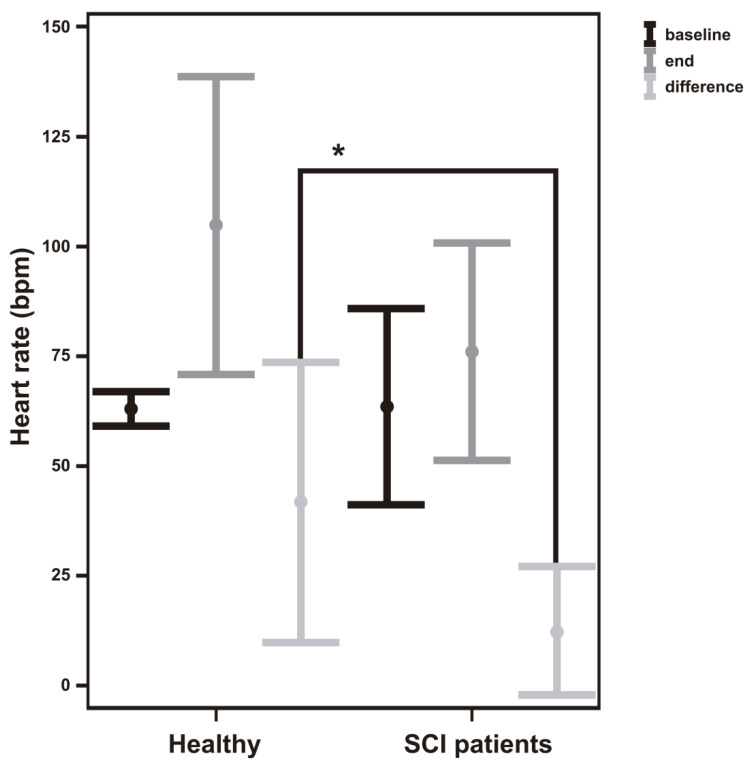
Mean heart rate measurements for neurologically healthy participants and SCI patients at baseline before the experimental session (black lines) and at the end of the experimental session (gray color), and the oscillation range (light gray color). * Significant statistically differences (*p* < 0.05).

**Figure 4 bioengineering-09-00763-f004:**
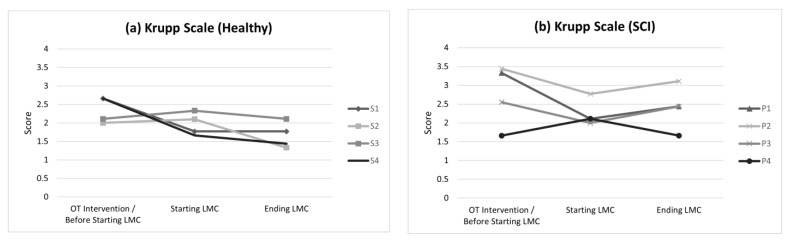
Evolution of the participants in the results obtained on the Krupp scale: (**a**) healthy participants and (**b**) SCI patients.

**Table 1 bioengineering-09-00763-t001:** Demographics and functional characteristics of the sample analyzed.

Variables	Sample Analyzed	
	Healthy (n = 4)	Patients (n = 4)
Sex(Male) *	1.00 ± 25.00	3.00 ± 75.00
Age(Years) +	31.50 ± 6.00	27.50 ± 13.20
Weight (kg)	62.00 ± 7.05	70.25 ± 7.25
Height (cm)	167.40 ± 10.24	172.50 ± 9.32
Etiology Injury (Traumatic)	-	4.00 ± 100.00
Time since injury (months)	-	5.50 ± 1.50
Injury Level		
C6	-	1.00 ± 25.00
C7	-	1.00 ± 25.00
C8	-	2.00 ± 50.00
AIS Classification		
A	-	2.00 ± 50
B	-	1.00 ± 25
C	-	-
D	-	1.00 ± 25
Right UE Motor score	25.00 ± 0.00 ^a^	20.50 ± 3.50 ^a^

^a^ (*p* < 0.01); * categorical variables are expressed as frequency and percentage; + continuous variables are expressed as mean and standard deviation.

**Table 2 bioengineering-09-00763-t002:** Outcome variables measured within the experimental session number 6 for healthy and SCI patients.

Variables	Healthy (n = 4)	SCI (n = 4)
Activities UL training (repetitions)		
Activity1 (Robot Assembly) *	24.50 ± 12.25 ^a^	11.00 ± 10.50 ^a^
Activity2 (Petal picking) *	80.00 ± 34.25 ^a^	26.50 ± 21.50 ^a^
Heart Rate (bpm)		
Baseline (before) +	63.50 ± 4.50	60,50±26.00
Ending session +	101.00 ± 38.25	78.00±36.00)
Difference (baseline-ending) +	38.50 ± 37.25 ^a^	9.00 ± 15.50 ^a^
Minimum +	53.00 ± 7.00	59.50 ± 9.50
Maximum +	149.50 ± 64.50 ^a^	100.50 ± 20.75 ^a^
Range +	92.50 ± 65.50 ^a^	44.00 ± 14.25 ^a^
Perceived effort		
Borg Scale (1–10)	3.00 ± 1.50	4.00 ± 1.50

a, *p* < 0.05. The results are expressed as mean and standard deviation, * this variable is expressed in repetitions achieved in the activity (rep.), + this variable is expressed in beats per minute (bpm).

**Table 3 bioengineering-09-00763-t003:** Results in the Krupp scale and the UL movements number at baseline (before starting the training) and at the end of all the experimental sessions.

	Baseline		Ending	
	Healthy (n = 4)	SCI (n = 4)	Healthy (n = 4)	SCI (n = 4)
Krupp Scale	2.33 ± 0.61	3.38 ± 1.28	1.99 ± 0.48	2.49 ± 0.69
UL Movement *	11.162 ± 5.055	1.418 ± 9.800	10.247 ± 6.739	12.409 ± 5.496

* The results are expressed as mean and standard deviation.

## Data Availability

Data sharing is not applicable to this article.

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
