# Peer review of "Is it Feasible to Use a Low-Cost Wearable Sensor for Heart Rate Monitoring within an Upper Limb Training in Spinal Cord Injured Patients?: A Pilot Study"

_bioengineering, 2022, doi:10.3390/bioengineering9120763_

Round 1
Reviewer 1 Report
Article title: Is feasible to use a low cost wearable sensor for heart rate monitoring within an upper limb training in Spinal Cord Injured patients?
# Overall statement or summary of the article:
The paper aims to analyze the feasibility of this wearable device for HR monitoring in cervical SCI patients by comparing the HR measured by means of XMB5 with those measured by a capno- graph/pulse oximeter used in the clinical setting for patients’ monitoring, simultaneously. The topic for this study is interesting; however, some major points are required before any progress.
# Since your study used a limited number of participants, please add “a pilot study " in the title.
# In section 1, please provide a paragraph at the end of the introduction starting with “The remainder of this paper consists of the following: Section 2 provides…..”
# please move the last sentence in the introduction (line 82-83) into Materials and Methods section
# In section 1, the authors should clearly mention the weak point of former works (identification of the gaps) and describe the novelties of the current investigation to justify that the paper deserves to be published in this journal. A comparative overview table that shows the key differences between the different previous methods and the proposed method should solve this point. Also, please consider using new references.
# Please use high-resolution and colorful images for figures 2, 3 and 4.
# The authors need to mention and discuss Figure 3 in the text.
# The discussion section should also contain what impacted the results, limitations and what is future work if any.
Author Response
Article title: Is feasible to use a low cost wearable sensor for heart rate monitoring within an upper limb training in Spinal Cord Injured patients?
Reviewer 1 (Round 1)
Reviewer: # Overall statement or summary of the article:
The paper aims to analyze the feasibility of this wearable device for HR monitoring in cervical SCI patients by comparing the HR measured by means of XMB5 with those measured by a capno- graph/pulse oximeter used in the clinical setting for patients’ monitoring, simultaneously. The topic for this study is interesting; however, some major points are required before any progress.
Authors: Thank you very much for the comments. We have been working very hard during this week for improving the manuscript.
Reviewer:# Since your study used a limited number of participants, please add “a pilot study " in the title.
Authors: Following your suggestion, we have included “a pilot study” in the manuscript title.
Reviewer:# In section 1, please provide a paragraph at the end of the introduction starting with “The remainder of this paper consists of the following: Section 2 provides…..”
Authors: Thank you very much for the suggestion. A paragraph at the end of the introduction section has been added in relation to the content of the manuscript.
Reviewer:# please move the last sentence in the introduction (line 82-83) into Materials and Methods section
Authors: This sentence was removed from the Introduction section and included in the “Participants” subsection within the “Materials and Methods” section.
Reviewer:# In section 1, the authors should clearly mention the weak point of former works (identification of the gaps) and describe the novelties of the current investigation to justify that the paper deserves to be published in this journal. A comparative overview table that shows the key differences between the different previous methods and the proposed method should solve this point. Also, please consider using new references.
Authors: Thank you very much for the comment. A paragraph including what we believe is new in this research has been included in the introduction section.
The truth is that these devices, which are commercially available, are being acquired for leisure and free time purposes, but it is becoming clear that they have great potential when applied to the clinical environment within biomedical applications in different fields. However, precisely because they are applied to patients, specifically neurological patients, we must take special care to study and to analyze the measurements obtained by means of these devices. For this reason, the Nonin LifeSense II capnograph and pulse oximeter (NLII) was chosen. This study has not previously been found in the literature.
Moreover, all the references have been revised and new references have been included.
Reviewer:# Please use high-resolution and colorful images for figures 2, 3 and 4.
Authors: Figures 2, 3 and 4 have been improved in resolution, and they are included in the text.
Reviewer:# The authors need to mention and discuss Figure 3 in the text.
Authors: Figure 3 is mentioned within the Results and Discussion sections.
Reviewer:# The discussion section should also contain what impacted the results, limitations and what is future work if any.
Authors: We appreciate your comment. To emphasise the potential impact of the results obtained, a paragraph has been included in the discussion. It also includes limitations and next steps for research in this field.
Reviewer 2 Report
This study discusses the feasibility of using a low cost wearable sensor for heart rate monitoring within specific conditions. This work contains some interesting points those are valuable to readers and peers working in this field. Some moderate revisions are required.
1. Refs: for the monitoring, related works like Si optical biosensor for environment monitoring and bio-sensing applications (see doi: 10.1364/OME.9.003985) using the Bi-CMOS technology could be included after refs. 5&6&13.
2. Technical impact: at the beginning of Section 2, one short summary presenting the connection between the following sub-sections should be given; 2.3. Equipments & 2.4. Experimental setup and data acquisition, the two sub-titles seem to b overlapped, could be combined effectively; at the end of 2.5, for the sensors, the one of silicon optical sensor (see doi: 10.1088/1361-6439/abf333) could be referred after ref. 23 as to enhance the technical impact of the study; for Section 3, one short summary should be given at the beginning as well; in Fig. 2, what the tendency of linearity is should be further clarified, also how to optimize the slope value toward better characteristics could be analyzed in detail; for the two pictures in Fig. 4, it is suggested to make the two together into one as to compare them directly and show the correlation.
3. Section 4 Discussion is unsatisfied, no more than two paragraphs would be fine, one for evaluating the study, one for further prospective; Section 5 Conclusion, one paragraph showing the most important highlight of the study would be fine.
THEREFORE, major revision is required before further consideration.
Author Response
Reviewer 2 (Round 1)
Reviewer:
This study discusses the feasibility of using a low cost wearable sensor for heart rate monitoring within specific conditions. This work contains some interesting points those are valuable to readers and peers working in this field. Some moderate revisions are required.
Authors: The authors are grateful for the comments and suggestions that have allowed us to improve the quality of the manuscript.
Reviewer:
- Refs: for the monitoring, related works like Si optical biosensor for environment monitoring and bio-sensing applications (see doi: 10.1364/OME.9.003985) using the Bi-CMOS technology could be included after refs. 5&6&13.
Authors: The suggested reference has been included in the text.
Reviewer:
- Technical impact: at the beginning of Section 2, one short summary presenting the connection between the following sub-sections should be given; 2.3. Equipments & 2.4. Experimental setup and data acquisition, the two sub-titles seem to b overlapped, could be combined effectively; at the end of 2.5, for the sensors, the one of silicon optical sensor (see doi: 10.1088/1361-6439/abf333) could be referred after ref. 23 as to enhance the technical impact of the study; for Section 3, one short summary should be given at the beginning as well; in Fig. 2, what the tendency of linearity is should be further clarified, also how to optimize the slope value toward better characteristics could be analyzed in detail; for the two pictures in Fig. 4, it is suggested to make the two together into one as to compare them directly and show the correlation.
Authors: A paragraph introducing the information within Materials and Methods has been included. Following the reviewer suggestion, subsections 2.3 and 2.4 were combined.
The reference proposed has been included in the text in “Materials and Methods” section, as reference 17.
A paragraph describing the results section has been included at the beginning of section 3.
Regarding the Figure 2, we believe that linearity cannot be expressed in any other way. Moreover, optimisation of the slope could only be done by increasing the experimental sample, which could be the subject of further research.
On the other hand, combining the two figures in Figure 4 into one would lead to more confusion. Therefore (we hope you will agree) we have matched the scales of the two figures to improve the understanding of the figures.
Reviewer:
- Section 4 Discussion is unsatisfied, no more than two paragraphs would be fine, one for evaluating the study, one for further prospective; Section 5 Conclusion, one paragraph showing the most important highlight of the study would be fine.
Authors: We believe that everything you request is included in the discussion. However, another reviewer has suggested to us that also the limitations of the study should be well described in the discussion. Therefore, we have not been able to limit it to the two paragraphs you request.
However, following your recommendation (thank you very much for this), we have revised the conclusion of the study.
Indeed, without your input we would not have been able to improve the paper.
Reviewer:
THEREFORE, major revision is required before further consideration.
Authors: Thank you very much for all the comments and suggestions for improving the paper.
Round 2
Reviewer 1 Report
This version of the manuscript has been significantly improved and the authors have answered most asked questions. I recommend the acceptance of the paper for publication.
Reviewer 2 Report
No further comments